# Antimicrobial Resistance and Antibiotic Consumption in a Secondary Care Hospital in Mexico

**DOI:** 10.3390/antibiotics13020178

**Published:** 2024-02-12

**Authors:** Elda Carolina Dávila-López, María Guadalupe Berumen-Lechuga, Carlos José Molina-Pérez, Rodolfo Norberto Jimenez-Juarez, Alfredo Leaños-Miranda, Natali Robles-Ordoñez, María Isabel Peña-Cano, German Alberto Venegas-Esquivel

**Affiliations:** 1Internal Medicine Departament, Hospital General Regional 251, Instituto Mexicano del Seguro Social (IMSS), Metepec 52148, Mexico; ec.davila@outlook.com; 2Medical Research Coordination, Órgano de Operación Administrativa Desconcentrada Regional Mexico Poniente, Instituto Mexicano del Seguro Social (IMSS), Toluca 50000, Mexico; 3Clinical Coordination, Hospital General de Zona 252, Instituto Mexicano del Seguro Social (IMSS), Atlacomulco 50454, Mexico; 4Infectious Diseases Departament, Federico Gómez Children’s Hospital of Mexico, Mexico City 06720, Mexico; 5Medical Research Unit in Reproductive Medicine, Unidad de Investigación Médica en Medicina Reproductiva, UMAE Hospital de Ginecología y Obstetricia 4 “Dr. Luis Castelazo Ayala”, Instituto Mexicano del Seguro Social (IMSS), Mexico City 01090, Mexico; alfredolm@yahoo.com; 6Infectious Diseases Department, Hospital para el Niño, Instituto Materno Infantil del Estado de Mexico, Toluca 50170, Mexico; 7Clinical Laboratory Department, Hospital de Ginecología y Obstetricia 221, Instituto Mexicano del Seguro Social (IMSS), Toluca 50150, Mexico; 8Pediatric Department, Hospital General Regional 251, Instituto Mexicano del Seguro Social (IMSS), Metepec 52148, Mexico

**Keywords:** antimicrobial drug resistance, anti-bacterial agents, antibiotic consumption, ESKAPE pathogens

## Abstract

Background: Antimicrobial resistance is a global health problem, due to morbidity, mortality, and healthcare costs. The misuse of antimicrobials is the main cause of antimicrobial resistance. The aim of this study was to report antimicrobial resistance and antibiotic consumption in a secondary care hospital in Mexico. Methods: Within a cross-sectional study, antimicrobial resistance data on ESKAPE pathogens (*Enterococcus faecium*, *Staphylococcus aureus*, *Klebsiella pneumoniae*, *Acinetobacter baumannii*, *Pseudomonas aeruginosa*, and *Enterobacter species*) and antibiotic consumption from 2020 to 2022 were collected. Antimicrobial resistance was reported based on percentages of resistance and consumption was analyzed using the defined daily dose (DDD)/100 bed days and the AWaRe (Access, Surveillance, Reservation) antibiotic group. Results: Antibiotic consumption in 2020, 2021 and 2022 was 330, 175 and 175 DDD/100 beds day, respectively. The rate of ceftriaxone resistance in *E. coli* (n = 526) and *K. pneumoniae* (n = 80) was 76% and 69%, respectively, the rate of carbapenem resistance in *A. baumannii* (n = 168) and *P. aeruginosa* (n = 108) was 92% and 52%, respectively; the rate of oxacillin resistance in *S. aureus* (n = 208) was 27%; and the rate of vancomycin resistance in *E. faecium* (n = 68) was 47%. Conclusion: The reported results are congruent with global estimates of antibiotic resistance and consumption, providing an overview that could generate actions for antimicrobial optimization at the local and regional levels.

## 1. Introduction

The treatment of infectious diseases with antibiotics has been beneficial for decades; however, their inappropriate or excessive use has promoted the development of antimicrobial resistance. The World Health Organization (WHO) has declared the rise of antimicrobial resistance to be a global health problem; it is now one of the 10 global health problems to be monitored and strengthened with actions to support stronger health systems [1,2,3,4].

Antimicrobial resistance is a process that occurs when microorganisms (such as bacteria, fungi, viruses, and parasites) undergo evolutionary changes that evade the mechanisms of action of antimicrobial drugs, enabling their survival [5]. 

AMR has clinical implications that impact the treatment of infections such as the spread of resistance mechanisms in community- and hospital-acquired infections, the failure of chemotherapy, organ transplants and surgeries where antibiotic treatment is fundamental to their the success, and increased hospital stays and healthcare costs [6].

The group of microorganisms known as “ESKAPE” (*Enterococcus faecium*, *Staphylococcus aureus*, *Klebsiella pneumoniae*, *Acinetobacter baumannii*, *Pseudomonas aeruginosa*, and *Enterobacter species*) are the leading cause of nosocomial infections worldwide [7]. ESKAPE pathogens are responsible for up to 40% of infections in hospitals, mainly in intensive care units, because they have multiple mechanisms of evasion of various antibiotic treatments. These infections cause high levels of mortality and high health costs, and currently, there are few therapeutic options to deal with them [8,9,10].

These resistance mechanisms are generated by various factors; some environmental, some agricultural, and some associated with animal and human health. In human healthcare, there are several important aspects that occur within hospitals, such as the lack of prevention and infection control, which favors the spread of these infections, generating outbreaks of multidrug-resistant infections; however, the major factor that can be considered the main cause of antibiotic resistance is the abuse of antibiotic treatments due to incorrect antibiotic prescriptions [5,10].

With these objectives in focus, antimicrobial stewardship programs (ASPs) should be present in each hospital with the main objective of promoting correct antibiotic prescriptions to reduce costs, optimize therapeutic outcomes and reduce antimicrobial resistance [11,12]. Appropriate metrics are needed to measure the quality, clinical impacts and financial impacts of ASP [13]. Antibiotic consumption measurements in ASPs allows for the surveillance of antibiotic prescriptions. The WHO classified antibiotics into three groups: the Access group of antibiotics, which have low resistance potential and are used for first-line or second-line therapies; the Watch group of antibiotics, used only with specific indications because of their higher resistance potential; and the Reserve group of antibiotics, which should be only used as a last resort when all other antibiotics have failed. In October 2019, the Access, Watch, Reserve (AWaRe) classification was updated and reformed as a classification database [13,14]. 

One of the strategies for the optimization and control of antibiotic prescribing is the creation of guidelines based on local microbiological data as well as the measurement of antibiotic consumption to facilitate effective surveillance [11,12]. In Mexico, there have been efforts to organize and implement ASPs, starting with antibiotic consumption measurement and antimicrobial resistance in order to evaluate the impact of possible interventions. An effective ASP should be led by an infectious disease physician, in conjunction with a team of antimicrobial specialists (such as a clinical pharmacist and a specialist in clinical microbiology), and with the support of medical direction to generate the strongest results [11]. One of the great challenges in Mexico is the construction of these multidisciplinary teams due to the lack of specific personnel for these functions, as well as the lack of local information to generate actions to improve antibiotic prescriptions. In 2018, a strategy focused on combating antibiotic resistance in Mexico emerged, and since then, various agreements have been generated in health institutions in Mexico that favor the surveillance of antibiotic resistance through the actions of ASPs [15,16]. 

The information in Mexico is limited; the largest studies that provide information on antibiotic resistance and consumption mostly comprise data from tertiary-level hospitals, which treat more complex diseases and also have ASP teams that monitor correct prescriptions and generate actions to improve the use of antibiotics; however, most hospital care in Mexico is provided in secondary-level hospitals, which are currently in the process of implementing ASP. Therefore, generating information in these units provides an overview of the interventions that will be carried out in these medical care units. The objective of this study was to measure antibiotic consumption and antibiotic resistance in a secondary care hospital of the Mexican Social Security Institute in the State of Mexico.

## 2. Results

### 2.1. Antimicrobial Resistance

Between July 2020 and December 2022, a total of 1188 isolates were analyzed; isolates were obtained from bronchial secretions, central and peripheral blood and urine samples, abscesses, tissue secretions, and peritoneal fluid specimens. *Escherichia coli* was identified in 526 isolates; *Staphylococcus aureus* was identified in 208 isolates; *Acinetobacter baumannii* was identified in 198 isolates; *Pseudomonas aeruginosa* was identified in 108 isolates; *Klebsiella pneumoniae* was identified in 80 isolates; and *Enterococcus faecium* was identified in 68 isolates.

Antimicrobial drug resistance for Gram-negative pathogens is reported in Table 1. It was divided by culture. The overall rate of antimicrobial resistance to ceftriaxone observed in *Escherichia coli* (n = 526) and *Klebsiella pneumoniae* (n = 80) was 76% and 69%, respectively, while the rate of resistance to carbapenem in *Acinetobacter baumannii* (n = 168) and *Pseudomonas aeruginosa* (n = 108) was 92% and 52%, respectively.

Antimicrobial drug resistance analysis for Gram-positive microorganisms is shown in Table 2. It was categorized by type of culture. The overall rate of oxacillin resistance for *Staphylococcus aureus* (n = 208) was 27% and 4% for vancomycin. The rate of antimicrobial drug resistance for vancomycin in *Enterococcus faecium* (n = 68) was 47%.

### 2.2. Antibiotic Consumption

In Table 3, the comparative analysis of antibiotic consumption by year and by hospital department is presented. The average antibiotic consumption in the hospital in 2020 was 330 DDD/per 100 beds days, for 2021 it was 174.69 DDD/100 bed days, and for 2022 it was 175 DDD/100 bed days. There was a significant decrease in 2021 and 2022 (*p* < 0.001). Antibiotic consumption was variable by both hospital and department.

The period observed is shown in Figure 1. The antibiotics with the highest rate of consumption in 2020 were clarithromycin, azithromycin, and ceftriaxone, with 87, 60 and 55 DDD/100 bed days, respectively; in 2021, the antibiotics with the highest rate of consumption were ceftriaxone, levofloxacin and clarithromycin, with 37, 27 and 22 DDD/100 bed days, respectively; and for 2022, the antibiotics with the highest rate of consumption were ceftriaxone, levofloxacin and metronidazole, with 38, 18 and 17 DDD/100 bed days, respectively.

The pattern of antibiotic consumption during the study period according to the WHO AWaRe categories is shown in Figure 2. The average antibiotic consumption rate in 2020 was 15.9%, 83.1% and 1.5% for the Access, Watch and Reserve groups, respectively. In 2021, the antibiotic consumption rate was 27.9%, 71.1% and 0.97% for the Access, Watch and Reserve groups, respectively, while in 2022, it was 35.3%, 63.87% and 0.77%, respectively. There was a significant reduction in antibiotic consumption in the Watch and Reserve groups in 2022 compared with 2021.

## 3. Discussion

In this cross-sectional study, we report on antimicrobial resistance and antibiotic consumption over a three-year period in a secondary care hospital in the State of Mexico in Mexico. It was observed that antibiotic consumption was higher in 2020, probably associated with the COVID-19 pandemic, as it was one of the units assigned to COVID-19 care in Mexico as well as general medical care. We also observed that antibiotics in the Watch and Reserve groups were used more frequently between July 2020 and April 2021; after the implementation of an ASP, there was a significant reduction in antibiotic use, and antibiotics in the Access group were used more frequently. These findings are consistent in the hospital overall and by department, with the exception of the intensive care unit, which displayed increased antibiotic consumption in 2022, probably associated with the knowledge of antibiotic resistance and hospital outbreaks. These results show greater resistance to third-generation cephalosporins as well as to carbapenems than previously reported.

Antimicrobial resistance is a public health problem. In 2019, an estimated 5 million deaths were associated with antimicrobial resistance, along with 1.27 million direct deaths [17,18]. Attention to this problem has motivated the elaboration of national action plans; in Mexico, there are reports from tertiary-care hospitals, but there is little information on secondary care hospitals. The Tracking Antimicrobial resistance Country Self-Assessment Survey (TrACSS) monitors the implementation of these antimicrobial resistance national action plans. The surveillance of antimicrobial resistance was under development in Mexico during 2022 [19]; this report provides local and regional information that allows for improving actions in the optimization of antimicrobials. However, there is still a long way to go for Mexico regarding antimicrobial stewardship measures.

Antibiotic resistance can be variable by region from country to country. Locally, there may be similarities in resistance patterns; however, it is preferable to have local data for actions within antibiotic optimization programs. Similar to our findings, in a systematic analysis of global antimicrobial resistance, it was reported that *Escherichia coli* isolates had resistance to third-generation cephalosporins at a rate of between 60 and 70%, *Acinetobacter baumannii* isolates presented resistance to carbapenem in 80% of cases, and *Staphylococcus aureus* had resistance to methicillin in 30–40% of cases [17,20]. In Latin America, in a hospital in Peru in 2018, the rate of carbapenem resistance in *Escherichia coli* was observed to be around 5%, for *Klebsiella pneumoniae*, it was observed to range from 27 to 31%, for *Pseudomonas aeruginosa*, it was observed to be 57% to 67%, and for *Acinetobacter baumannii*, it was found to be higher than 85%. In the case of *Staphylococcus aureus*, the rate of methicillin resistance was found to be around 54% to 83%, and in the case of *Enterococcus faecium*, the rate of vancomycin resistance was found to be 58% to 60% [10]. In Mexico in 2018, the Universidad Nacional Autónoma de Mexico (UNAM), through the Plan Universitario de Control de la Resistencia Antimicrobiana (PUCRA), reported that the rate of resistance to third-generation cephalosporins was 62% for *Escherichia coli* and 31.1% for *Klebsiella pneumoniae*, while the rate of resistance to carbapenems was 25% for *Pseudomona aeruginosa* and 64% for *Acinetobacter baumannii*. *Staphylococcus aureus* showed resistance to oxacillin in 25% of isolates and resistance to vancomycin was observed in 36% of *Enterococcus faecium* isolates [21]. In 2019, the INVIFAR group published a cumulative study on 47 hospitals in Mexico, reporting resistance to third-generation cephalosporins in 50.9% of *Escherichia coli* isolates and 31.1% of *Klebsiella* spp. isolates; resistance to carbapenems in 27.8% of *Pseudomona aeruginosa* isolates and 79.6% of *Acinetobacter baumannii* isolates; resistance to oxacillin in 23.1% of *Staphylococcus aureus* isolates; and resistance to vancomycin in 20% of *Enterococcus faecium* isolates [22].

Although we do not know how high antibiotic resistance rates were prior to the COVID-19 pandemic in this hospital due to the absence of data, we can hypothesize that there was an increase in bacterial resistance due to an increase in antibiotic consumption given the lack of awareness of the disease, which was evident in studies reflecting post-pandemic antibiotic consumption [23]. Our results show variability in antibiotic consumption in 2020 associated in temporality with the COVID-19 pandemic. Similar findings have been described by other authors. Fukushiage and collaborators reported in a systematic review an increase from 10% to 20% in antibiotic consumption [24], while a four-fold increase in antibiotic consumption during the COVID-19 pandemic was also reported in a systematic review [25]. In 2020, Ponce and collaborators reported the antibiotic consumption of 20 hospitals in Mexico from 2016 to 2017, being between 20 and 95 DDD/100 beds per day; meanwhile, the PUCRA network reported ranges from 20 to 90 DDD/100 beds per day, predominantly with the consumption of cephalosporin in both reports [21]. The Jordan Food and Drug Administration also reported an increase in the consumption of certain antibiotics, mainly third-generation cephalosporins, increasing by 19%, macrolides, increasing by 52%, carbapenems, increasing by 52%, and lincomsamides, increasing by 106%, during the COVID-19 pandemic [23]. This is similar to our results, where the antibiotics with the highest rates of consumption were cephalosporins, quinolones and macrolides during 2020, the year in which the hospital had the highest number of COVID-19 cases.

An antibiotic optimization program was initiated as a response to the increase in healthcare-associated infections in patients with COVID-19, mainly due to outbreaks of ventilator-associated pneumonia, mainly due to Acinetobacter baumannii, which was one of the main microorganisms coinfecting patients with severe COVID-19 disease [26]. With the start of vaccination in Mexico (December 2020) and a decrease in hospitalization at the national level, there was a reduction in antibiotic consumption by up to 50% [24,27]. However, the success in reducing antibiotic consumption cannot be entirely attributed to the implementation of an ASP, as it is also explained by the decrease in COVID-19 cases. However, by 2021, there was an improvement in prescribing by increasing the prescription of antibiotics from the Access group. The presence of healthcare-associated infections, the increase in antibiotic resistance, and the lack of correct antibiotic use in hospitals should be a call to health authorities to organize and promote ASPs, which in multiple studies have been shown to lead to a reduction in antibiotic consumption [11]. Mexico has at least three different health systems, among which prescribing practices may vary due to drug availability and the type of medical care, which explains the differences exhibited in the antimicrobial resistance surveillance networks [21].

In our report, an increase in antibiotic consumption during the COVID-19 pandemic was evident, mainly with the consumption of cephalosporins, quinolones, and macrolides. There was no congruence between antibiotic consumption and antimicrobial resistance, as was observed with the use of quinolones and resistance to this drug, despite the fact that consumption should have decreased due to their adverse effects. This information reinforces the need for action in the optimization of antibiotics in Mexico [1,21].

Among the limitations observed in our study is the lack of patient data, especially regarding clinical diagnosis, which would provide more strength for future research to differentiate between community- and hospital-acquired infections, favoring better decision making by ASPs. Another weakness found was the variation in the length of hospital stays due to the COVID-19 pandemic, which affected antibiotic consumption trends, because during the COVID-19 pandemic, the hospital underwent different modifications to promote the care of patients with COVID-19. The presence of hospital outbreaks during the COVID-19 pandemic, as previously mentioned, mainly related to ventilator-associated pneumonia due to *Acinetobacter baumannii*, changed the level of antibiotic consumption, especially in intensive care areas. These data are relevant to consider in future studies where dates are specified to justify alterations in antibiotic consumption. The final limitation to consider is the lack of microbiological data from before the pandemic, meaning that we were unable to perform a comparative study to assess the impact on antibiotic resistance secondary to the increase in antibiotic consumption during the COVID-19 pandemic.

Regardless of the limitations of this study, this information provides a scenario of antibiotic resistance and consumption in a secondary-level hospital in Mexico, representing the main type of hospital for most of the population in Mexico, revealing the need for ASP teams in these units to initiate actions against antibiotic resistance as part of the actions established worldwide.

## 4. Materials and Methods

A cross-sectional study was conducted at the Hospital General Regional 251 of the Instituto Mexicano del Seguro Social. The Instituto Mexicano del Seguro Social is one of the main organizations providing healthcare in Mexico. It is composed of different hospitals of all levels throughout the country; however, it mostly comprises secondary-level care units. The Hospital General Regional 251, located in the State of Mexico in a town called Metepec, is a secondary care unit with 262 beds; it provides internal medicine, surgery, intensive care unit, gynecology and obstetrics, and pediatrics services, and therefore, it provides care to adult and pediatric patients with diverse diseases, including patients with cancer, renal insufficiency, immunological diseases, and other conditions. During the COVID-19 pandemic, this hospital was restructured by delimiting spaces for COVID-19 patients and for general care at a lower percentage. In 2020, an infectious diseases specialist started an antimicrobial stewardship program with the following interventions: auditing antibiotic therapy; providing feedback to healthcare professionals; measuring bacterial resistance; monitoring antibiotic consumption; and implementing an educational training program for medical residents of internal medicine, in which cultures were taken before the use of antibiotics. 

### 4.1. Microbiological Analysis

The BactAlert^®^ platform was used for blood culture incubation. Cultures of bronchial secretion, abscesses and secretions, urine and peritoneal fluid were performed in standard way. Genus and species identification, as well as the identification of antibiotic sensitivity, were performed with the automatized system PhoenixBD^®^. We only included ESKAPE pathogen strains from November 2021 to December 2022 by recording the sensitivity and resistance provided in the automated system report.

Antibiotic consumption was obtained from the pharmacy registry. The length of hospital stay was established based on the medical information system from July 2020 to December 2022. For the consumption analysis, DDD/100 bed days was used, using the AMC Tool software version 1.9.0. Consumption was also assessed according to the AWaRe classification proposed by WHO. Pediatric care areas were excluded because they required a different antibiotic consumption measurement. 

### 4.2. Statistcal Analyses

The rate of bacterial resistance was analyzed in general for all cultures, and for each type of culture (blood, urine, bronchial secretion, abscesses and/or secretions and peritoneal fluid), adjusting the percentage of resistance. Cultures with the same species obtained in the same day were excluded. 

Antibiotic consumption was assessed as the rate of DDD/100 bed days for each antibiotic in the whole hospital and in every department. Additionally, we reported the monthly percentage of antibiotic consumption according to the AWaRe classification. Once the data were collected, they were compared by year (2020, 2021 and 2022) for the whole hospital and for every department. 

Normality and variance homogeneity tests were performed to determine parametric and nonparametric values. One-way ANOVA and Kruskal–Wallis tests were applied to compare antibiotic consumption according to the data distribution. A 2-tailed *p*-value < 0.05 was considered statistically significant; data were analyzed using SPSS statistical software version 27.

## 5. Conclusions

These results provide information on pathogen resistance and antibiotic consumptions in Mexico. This is a likely scenario for secondary care hospitals, which reinforces the need to organize antimicrobial stewardship programs to generate local guidelines, monitor, audit, and provide feedback about antibiotic consumption, and implement educational interventions for healthcare professionals.

## Figures and Tables

**Figure 1 antibiotics-13-00178-f001:**
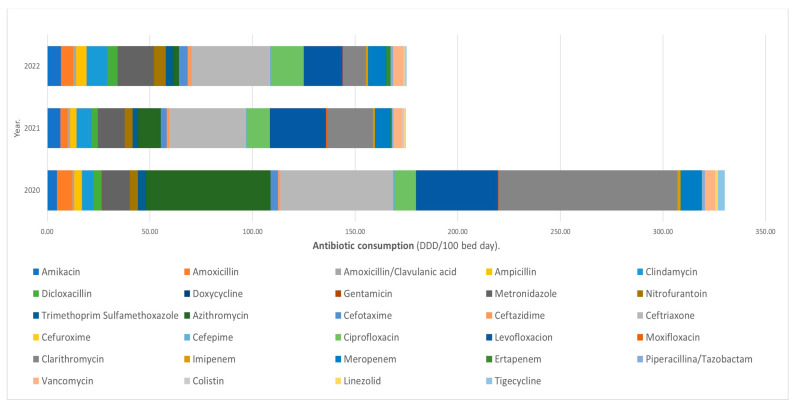
Proportion of antibiotic consumption per DDD/100 bed day per year at Hospital General Regional No. 251, Metepec, Mexico.

**Figure 2 antibiotics-13-00178-f002:**
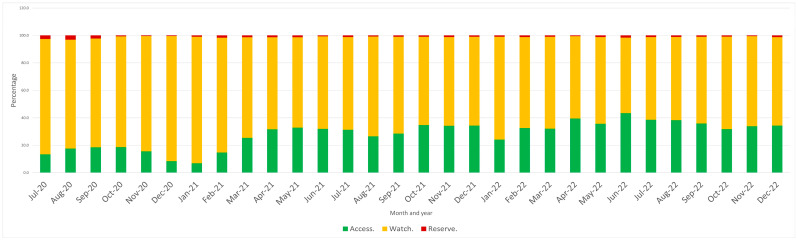
Proportion of antibiotics used monthly according to the AWaRe classification at Hospital General Regional 251, Metepec, Mexico.

**Table 1 antibiotics-13-00178-t001:** Rates of antibiotic resistance by culture of Gram-negative microorganisms belonging to the ESKAPE group from a secondary care hospital in Mexico.

Culture	Microorganisms	Isolations (n)	Antimicrobial Drug Resistance (%)											
			AMP	AMK	CAZ	CIP	CRO	FEP	IMP	MEM	SAM	TZP	ETP	SXT
Bronchial secretions	*A. baumannii*	60	ND	93	95	93	ND	93	93	92	95	97	ND	92
	*A. baumannii calcoaceticus*	11	ND	82	82	82	ND	82	82	82	91	82	ND	82
	*E. coli*	18	78	0	72	83	78	72	17	6	67	22	17	44
	*K. pneumoniae*	14	93	7	36	36	36	36	14	14	64	29	14	36
	*P. aeruginosa*	19	ND	32	63	53	ND	68	68	68	ND	63	ND	ND
Central blood	*A. baumannii*	25	ND	96	100	96	ND	96	96	96	96	92	ND	92
	*A. baumannii calcoaceticus*	2	ND	50	50	50	ND	50	50	50	50	50	ND	50
	*A. baumannii lwoffi*	1	ND	100	100	100	ND	100	100	100	0	0	ND	100
	*E. coli*	26	81	0	54	77	73	69	0	0	81	15	0	42
	*K. pneumoniae*	15	93	0	47	87	80	60	7	0	80	27	13	67
	*P. aeruginosa*	15	ND	27	40	33	ND	40	53	53	ND	40	ND	ND
Peripheral blood	*A. baumannii*	17	ND	76	82	76	ND	76	76	76	94	82	ND	71
	*A. baumannii calcoaceticus*	1	ND	100	100	100	ND	100	100	100	100	100	ND	100
	*E. coli*	19	84	0	68	84	74	68	5	11	89	26	11	42
	*K. pneumoniae*	4	100	0	100	100	100	100	0	0	100	50	0	100
	*P. aeruginosa*	6	ND	50	83	50	ND	67	83	67	ND	83	ND	ND
Abscesses and other secretions	*A. baumannii*	19	ND	95	95	95	ND	95	95	95	100	95	ND	95
	*A. baumannii calcoaceticus*	2	ND	100	100	100	ND	100	100	100	100	100	ND	100
	*E. coli*	107	93	1	59	89	81	72	15	14	95	31	17	67
	*K. pneumoniae*	5	100	0	80	80	80	80	20	20	100	20	20	60
	*P. aeruginosa*	18	ND	44	56	61	ND	56	50	56	ND	56	ND	ND
Urine samples	*A. baumannii*	23	ND	96	100	100	ND	100	96	96	100	96	ND	91
	*A. baumannii calcoaceticus*	7	ND	100	100	100	ND	100	100	100	100	100	ND	100
	*A. baumannii lwoffi*	2	ND	50	100	100	ND	100	100	50	100	100	ND	50
	*E. coli*	282	89	4	65	89	75	71	11	10	87	27	13	63
	*K. pneumoniae*	19	100	5	68	68	68	68	11	16	79	58	21	68
	*P. aeruginosa*	19	ND	68	74	79	ND	74	68	68	ND	79	ND	ND
Peritoneal fluids	*A. baumannii*	5	ND	100	100	100	ND	100	100	100	100	100	ND	80
	*E. coli*	19	89	11	68	89	68	68	16	5	74	16	11	74
	*K. pneumoniae*	8	88	25	75	75	88	88	38	25	88	50	50	63
	*P. aeruginosa*	13	ND	15	15	15	ND	23	23	15	ND	23	ND	ND
Total	*A. baumannii*	168	ND	92	95	93	ND	93	92	92	96	93	ND	89
	*A. baumannii calcoaceticus*	26	ND	88	88	88	ND	88	88	88	92	88	ND	88
	*A. baumannii lwoffi*	4	ND	75	100	100	ND	100	100	75	75	75	ND	75
	*E. coli*	526	89	4	63	87	76	71	12	11	86	27	14	63
	*K. pneumoniae*	80	96	10	60	69	69	64	16	15	78	40	21	63
	*P. aeruginosa*	108	ND	42	57	53	ND	58	60	57	ND	58	ND	ND

Results are expressed as a percentage of antimicrobial drug resistance accordingly to Phoenix BD^®^. Report. AMP, ampicillin; AMK, amikacin; CAZ, ceftazidime; CIP, ciprofloxacin; CRO, ceftriaxone; FEP, cefepime; IMP, imipenem; MEM, meropenem; SAM, ampicillin-sulbactam; TZP, piperacillin-tazobactam; ETP, ertapenem; SXT, trimethoprim sulfamethoxazole; ND, none determined.

**Table 2 antibiotics-13-00178-t002:** Rate of antibiotic resistance by culture of Gram-positive microorganisms belonging to the ESKAPE group from a secondary care hospital in Mexico.

Culture	Microorganisms	Isolations (n)	Antimicrobial Drug Resistance (%)									
			AMP	CLI	CIP	DAP	GEN	LZD	OXA	PEN	SXT	VAN
Bronchial secretions	*E. faecalis*	7	0	ND	100	14	57	0	ND	0	ND	14
	*E. faecium*	2	50	ND	100	0	50	0	ND	0	ND	50
	*S. aureus*	21	100	48	ND	0	5	0	48	76	5	0
Central blood	*E. faecium*	18	83	ND	100	0	39	0	ND	67	ND	39
	*S. aureus*	51	90	31	ND	2	14	0	29	84	4	6
Peripheral blood	*E. faecium*	6	67	ND	100	0	33	0	ND	67	ND	50
	*S. aureus*	44	93	30	ND	0	7	0	30	91	0	100
Abscesses and other secretions	*E. faecium*	9	89	ND	100	0	44	0	ND	89	ND	78
	*S. aureus*	42	98	17	ND	0	17	0	10	86	5	2
Urine sample	*E. faecium*	20	95	ND	95	0	30	0	ND	100	ND	40
	*S. aureus*	1	100	100	ND	100	0	0	100	100	0	0
Peritoneal fluids	*E. faecium*	4	100	ND	100	0	25	0	ND	75	ND	75
	*S. aureus*	12	83	25	ND	8	8	8	25	83	8	8
Total	*E. faecium*	68	85	ND	99	0	35	0	ND	79	ND	47
	*S. aureus*	208	93	31	ND	2	12	1	27	85	5	4

Results are expressed as a percentage of antimicrobial drug resistance accordingly to Phoenix BD^®^. report. AMP, ampicillin; CLI, clindamycin; CIP, ciprofloxacin; DAP, daptomycin; GEN, gentamicin; LZD, linezolid; OXA, oxacillin; PEN, penicillin; SXT, trimethoprim sulfamethoxazole; VAN, vancomycin.

**Table 3 antibiotics-13-00178-t003:** Comparative analysis of antibiotic consumption per DDD/100 bed day per year and per hospital department from a secondary care hospital in Mexico.

Service	Year	Average Length of Hospital Stays. (%)	Antibiotic Consumption Mean (DDD/100 Bed Days)	SD	*p* Value
Hospital	2020	68	330.05	99.20	0.001
	2021	83	174.69	75.34	
	2022	69	175.00	19.9	
Internal Medicine	2020	100	241.55	28.70	0.001
	2021	94	94.48	62.63	
	2022	73	92.32	13.41	
Surgery	2020	46	547.75	289.69	0.001
	2021	85	140.65	82.81	
	2022	88	124.42	20.45	
Gynecology and obstetrics	2020	12	254.22	92.34	0.001
	2021	26	190.29	91.89	
	2022	58	97.92	19.01	
Intensive Care Unit	2020	68	69.29	57.93	0.006
	2021	74	40.28	58.40	
	2022	73	119.03	36.06	

DDD, defined daily dose; SD, standard deviation.

## Data Availability

Data that support the findings of this study are available from the corresponding author upon reasonable request.

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
