# Peer review of "Antimicrobial Resistance and Antibiotic Consumption in a Secondary Care Hospital in Mexico"

_antibiotics, 2024, doi:10.3390/antibiotics13020178_

Round 1
Reviewer 1 Report
Comments and Suggestions for Authors
The present study delves into one of the most salient health topics, i.e., antimicrobial resistance and antibiotic consumption, with a specific focus on a secondary care unit in Mexico. The topic holds significance due to the global rise in antibiotic resistance, and this paper's outcomes can contribute valuable data to this area, expanding existing knowledge.
Strengths:
1. The study addresses a highly relevant topic in a specific geographic context, contributing empirical data to a field that requires such knowledge to formulate effective public health strategies and interventions. This can be crucial for future policy-making and health planning.
2. The methodology employed in the paper is robust. By conducting a cross-sectional study and employing the Defined Daily Dose as a standard metric, the researchers have followed an established and commonly used practice, which makes the findings comparable to other studies.
3. The focus on ESKAPE pathogens and their resistance patterns provides valuable insights. The detailed reporting of the resistance rates of different drugs for various strains of these pathogens is commendable.
Areas to Improve:
1. The paper seems to lack a detailed and comprehensive literature review section, bringing together existing knowledge in the field. This would help to better contextualize the study, highlighting its novelty and significance better.
2. The discussion on the limitation of the study is helpful, but the authors could consider elaborating more on how these limitations may have influenced the results and how they could be mitigated in future research.
3. It would be informative to include more background on the specific setting (i.e., the secondary care unit) in which the study was conducted. Details such as the demographics of patients, healthcare practices, and local antibiotic prescription guidelines could influence the results.
4. While the statistical analysis used is applicable for the data, it is not clear whether the authors checked for any assumptions of these tests (normality, homogeneity of variance, etc.). Further clarification would improve the validity of the paper.
5. The data on antibiotic consumption seems to be represented as aggregated numbers for different years. Breaking this data down into more specific time points (monthly/quarterly), if possible, could provide more detailed insights.
6. In future investigations, it would be interesting to further analyze the connection between certain demographic or clinical factors (such as age, sex, or comorbidities) with antibiotic resistance patterns. This could help to identify possible risk groups and adjust treatment strategies accordingly.
7. In the tables, the abbreviation dot is missing after each genus name.
Overall, the paper presents useful findings related.
Comments on the Quality of English Language
The language used is appropriate for scientific communication.
Author Response
Please see attached file in relation to reviewer 1 with data highlighted in green.

Reviewer 2 Report
Comments and Suggestions for Authors
Comment to authors
- Abstract: This section need not be structured nor numbered, in conformity with the journal style
- Lines 25-26: Responsible use/consumption of antimicrobials is NOT a global health problem. So, delete “and antibiotic consumption” in line 25 and revise the sentence accordingly
- Lines 25-26: No need to bold the aim of the study
- Line 28: “A cross-sectional study design ….”?
- Line 28: What is ESKAPE? Acronyms must be defined at their first use in a manuscript and then consistently used afterwards throughout the paper
- Lines 31-32: This sentence is misleading. It gives the impression that the authors categorised antibiotics into the AWaRe group. Re-right to reflect that “Exposure to antibiotics was reported according to the Access, Watch, and Reserve
(AWaRe) framework”
- Lines 33-37: Needs to be revised for clarity. Which of the third-generation cephalosporin antimicrobial agent(s) were the authors referring to?
- There is a need to provide a context and definitions for terms like “Antimicrobial resistance”. Antimicrobial stewardship”, etc in the introduction
- Lines 47-48: citations 1-2, are not that of WHO as claimed by the authors. The authors may wish to cite the World Health Organisation (2021). Top 10 global health issues to track in 2021. Available at: https://www.who.int/news-room/spotlight/10-global-health-issues-to-track-in-2021. Again although the manuscript, the citation numbers were written in round brackets ( ) instead of square brackets [ ], in line with the journal style
- Line 59: Why write “antimicrobial stewardship programme” in full having provided the acronym earlier in line 57?
- Line 71: “…be led by infectious disease physicians”?
- Line 88: Comment number 7 applies
- Tables 1-3: The table titles are so brief that they cannot stand alone. Table titles should be concise but informative enough to facilitate the full understanding of the table without referring to the body of the manuscript
- Line 205-206: Why then did the authors provide AMR results from 2020 to 2022?
- Line 218: Unnecessary repetition of an already-defined acronym
- Materials and method: What guided the choice/selection of antimicrobial agents used in the AMR/sensitivity testing?
- The study did not provide recommendations to guide prudent antimicrobial usage and limit AMR in Mexico
- Informed Consent Statement: Authors should provide certified evidence of the exception, by the hospital management or any other appropriate authority, as claimed in the manuscript.
Moderate English language revision is required for the clarity of the manuscript
Author Response
Please see attached file in relation to reviewer 2 with data highlighted in yellow.

Round 2
Reviewer 1 Report
Comments and Suggestions for Authors
The authors have made a number of corrections to the manuscript, which I believe is now acceptable for publication.
Author Response
We are grateful for your recommendations and comments please see the attachment.

Reviewer 2 Report
Comments and Suggestions for Authors
Comments to authors
The authors have improved the manuscript but I have a few further comments as follows:
- Line 26: Replace “overuse” with “misuse”. It is misuse (under-dose, incomplete dosage, wrong usage, etc) of antimicrobials that contribute mostly to AMR
- Lines 50-56. The sentence is rather too long. Should be fragmented into two or more sentences, with appropriate citations provided, to be reader-friendly. The definition of AMR provided by the authors suggests limited knowledge of the subject. The authors may wish to understand that not all cases of non-susceptibility of microbial pathogens to antimicrobial agents are AMR. There is the phenomenon of non-susceptibility to antimicrobials called Persistence. The definition provided did not discriminate between AMR and Persistence
- Table 1 title and others: Percentages of antibiotic-resistant Gram-negative bacteria, belonging to the ESKAPE group, from a general hospital in Mexico?
- Lines 249-266: Limitations of the study should be separated from the discussion. It should come before the reference list under the heading “Limitations of the study”
- Lines 268-273: You can provide the coordinates of the Hospital General Regional 251
- References not listed according to the journal style
Author Response
We are grateful for your recommendations and comments please see the attachment
